# Evaluation of the Xpert MTB/XDR test for detection of isoniazid, fluoroquinolones, and second-line injectable drugs resistance to *Mycobacterium tuberculosis*—Anhui Province, China

Ruiqing Zhang[1☯], Xundi Bao[2☯], Fangjin Bao[3, ☯], Chong Teng[4], Dongfang Xu[2], Zhou Liu[5], Yue Li[5], Bing Zhao[1], Hui Xia[1], Ruida Xing[1], Xichao Ou[1]*, Yanlin Zhao[1]*

1 National Key Laboratory of Intelligent Tracking and Forecasting for Infectious Diseases, National Center for Tuberculosis Control and Prevention, Chinese Centre for Disease Control and Prevention, Beijing, China, 2 Microbiological Institute, Anhui Provincial Center for Disease Control and Prevention, Hefei, China, 3 Tuberculosis Prevention and Control Institute, Anhui Provincial Center for Disease Control and Prevention, Hefei, China, 4 Department of Tuberculosis, Beijing Dongcheng District Center for Disease Control and Prevention, Beijing, China, 5 Department of Clinical Laboratory, Anhui Chest Hospital, Hefei, China

☯ These authors contributed equally to this work.
* ouxc@chinacdc.cn (OXC); zhaoyl@chinacdc.cn (ZYL)

## Abstract

### Introduction

The emergence of drug-resistant tuberculosis (DR-TB) has posed significant challenges to TB control. This study assessed the diagnostic performance of the Xpert MTB/XDR test for detecting drug resistance in TB patients.

### Methods

This study analyzed 276 samples collected from clinically suspected MDR-TB patients in Auhui Chest Hospital from 01/03/2022–01/03/2023. The Xpert MTB/XDR test was evaluated for its ability to detect resistance to isoniazid (INH), ethionamide (ETH), fluoroquinolones (FLQ), and second-line injectable drugs (SLIDs) compared with phenotypic drug susceptibility testing (pDST). Specimens were investigated by Sanger sequencing, where the MTB/XDR test and pDST results were inconsistent. Afterward, the clinical performance of the Xpert MTB/XDR test was also evaluated with the composite reference test (pDST+sequencing).

### Results

The sensitivity of the Xpert MTB/XDR test against pDST in detecting resistance to INH and FLQ using 276 samples was 95.77% (95% CI: 91.83–98.16) and 93.83% (95% CI: 86.18–97.97), respectively. In contrast, a lower sensitivity of the MTB XDR

**Data availability statement:** All relevant data are within the paper and its Supporting information files.

**Funding:** This work was supported by the National Key R&D Program of China (No. 2022YFC2305204, 2023YFC2307301), Cepheid Investigator-Initiated Study award (Cepheid-IIS-2023-0001), and Public Health Personnel Training Sopport Program (01056). The funders did not have a role in the study design, data collection, analysis and interpretation, and manuscript preparation.

**Competing interests:** All authors disclose no conflicts of interest.

test in predicting SLIDs and ETH resistance (sensitivity < 75%) compared with pDST was demonstrated in this study. The specificity for detecting all drugs was greater than 90%. Thirty-three samples were retested by sequencing, which identified mutations predicting INH and FLQ resistance, determining whether resistant or not by combining pDST and sequencing results. When considering pDST + sequencing, the sensitivity and specificity of the MTB/XDR assay for INH and FLQ drug targets increased, especially the detection specificity of FLQ has reached 100% (95% CI: 97.95–100).

## Conclusion

The Xpert MTB/XDR has high sensitivity and specificity in drug-resistant tuberculosis patients, making it better suited to meet the needs of rapid, sensitive, and accurate detection for drug-resistant tuberculosis in resource-limited settings, and serving as a critical tool for achieving personalized treatment and TB control.

## Introduction

In recent years, the emergence of drug-resistant tuberculosis (DR-TB), especially multidrug-resistant tuberculosis (MDR-TB) and extensively drug-resistant tuberculosis (XDR-TB), has posed significant challenges to TB control. In 2023, China estimated 741,000 new TB cases and 29,000 multidrug-resistant or rifampicin-resistant tuberculosis (MDR/RR-TB) cases [1]. However, only 40% of the projected MDR/RR-TB patients were confirmed by laboratory testing.

WHO recommends routine testing for resistance to Rifampicin (RIF), isoniazid (INH), and fluoroquinolones (FLQ) in all TB patients [2]. Phenotypic drug susceptibility test (pDST) could simultaneously detect first-line and second-line anti-tuberculosis drug resistance to *M. tuberculosis,* serving as the gold standard for diagnosing MDR/RR-TB. However, its disadvantages include cumbersome operation, a longer detection period, and high biosafety requirements.

Recently, nucleic acid amplification methods (NAATs) have been progressively used to diagnose MDR/RR-TB. Among them, in 2017, the WHO recommended Xpert MTB/RIF Ultra for the simultaneous detection of *Mycobacterium Tuberculosis* (MTB) and RIF resistance from patient samples. Compared to the Xpert MTB/RIF, the Xpert MTB/RIF Ultra demonstrates higher sensitivity in detecting samples with low bacterial loads, including those from extrapulmonary tuberculosis, AIDS-complicated tuberculosis, and pediatric tuberculosis [3,4]. In addition, the Xpert MTB/XDR test is used for rapid and prolonged drug resistance profiling for all TB-positive cases to identify MDR/pre-XDR TB, allowing for more precise treatment planning [5,6]. The Xpert MTB/XDR test enables rapid molecular DST within 90 minutes. When combined with the frontline Xpert MTB/RIF Ultra, the Xpert MTB/XDR test is a nested real-time PCR test with 10-color modules that set new standards in detecting mutations associated with resistance to isoniazid (INH), ethionamide (ETH), fluoroquinolones (FLQ), and second-line injectable drugs (SLIDs) in a single test. To detect INH resistance, four probe targets are inhA promoter, katG, fabG1, and oxyR-ahpC; for ETH resistance,

one probe target is inhA promoter; for FLQ resistance, three probes target gyrA QRDR and one target gyrB QRDR; for SLIDs resistance (AMK, KAN, CAP), one probe targets *rrs* and another targets is promoter. The Xpert MTB/XDR test can detect INH and FLQ resistance at low levels, as well as individual versus cross-resistance to SLID [7].

We reported the evaluation findings of the Xpert MTB/XDR test for the detection of INH, ETH, FLQ, and SLIDs resistance to *Mycobacterium tuberculosis*, completed at Anhui Chest Hospital between November 2022 and December 2023.

## Materials and methods

### Specimen

This study enrolled 289 samples sourced from clinically suspected MDR-TB patients between 01/03/2022 and 01/03/2023. These samples were stored at Anhui Provincial Chest Hospital, and we obtained the samples along with the patients' identification numbers on 01/03/2023. Clinically suspected MDR-TB patients have the following characteristics: ①Treatment failure with standard first-line anti-tuberculosis drugs (e.g., isoniazid and rifampicin); ②Active pulmonary lesions detected on chest CT scans; ③Persistent typical tuberculosis symptoms;④Positive sputum culture or molecular biological detection [8,9]. Of these, 13 samples that tested negative for culture and MTB were excluded. Furthermore, 276 culture-positive and MTB-positive samples had well-defined information on Culture, Smear, MTB identification, and Xpert MTB/RIF Ultra (Cepheid, USA) results. The details of the aforementioned approaches are provided in Ref (Fig 1). The study was conducted in accordance with the Declaration of Helsinki and approved by the Institutional Review Board of China CDC (202201).

### Xpert MTB/XDR test and phenotypic drug susceptibility test

To process pus or feces samples, add 1–2 mL of sterile PBS and mix thoroughly. Centrifuge at 3,000×g for 20 minutes and discard the supernatant. For lavage or puncture fluids, centrifuge at 3,000×g for 15 minutes and discard the

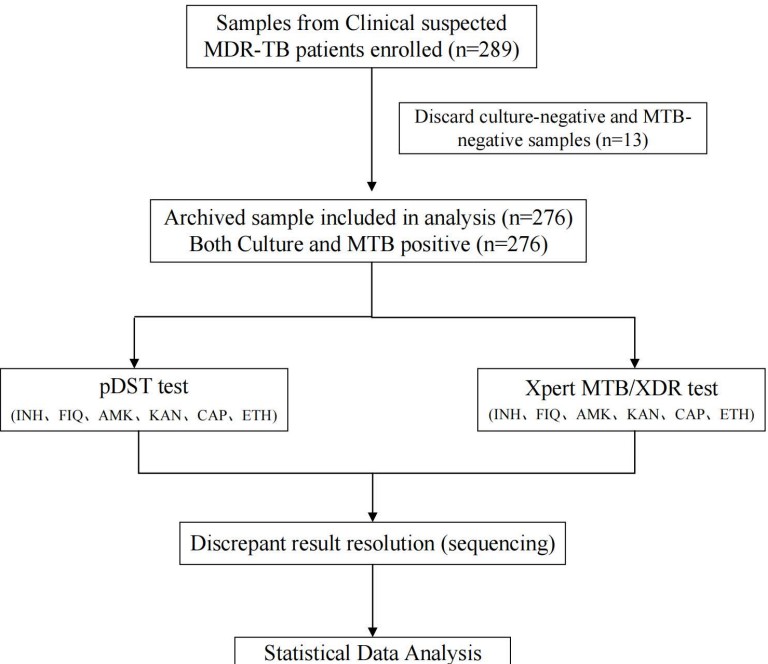

**Fig 1. The flow diagram for evaluating the Xpert MTB/XDR test for detection of Isoniazid, Fluoroquinolones, and second-line injectable drugs resistance to Mycobacterium tuberculosis.**

supernatant. Then, transfer 1 mL of sterile PBS solution to resuspend the sediments. Subsequently, add 2 mL of the Sample Reagent (SR) to 1 mL of unprocessed sputum or the liquid following the previous treatment. Then, the SR-sample mixture should be homogenized at room temperature by vigorous shaking 10–20 times and incubated for 10 minutes before being shaken again 10–20 times and incubated for another 5 minutes. The SR-sample combination was then tested using the Xpert MTB/XDR system, following the manufacturer's recommendations. If specimen testing yields a result of "ERROR," "INVALID," or "NO RESULT", the specimen will be re-tested if enough specimens are available. If the re-test result was also non-determinant, the specimen was reported as such.

Phenotypic DST utilized culture-positive pure strains of M. tuberculosis. To create a 1 mg/mL (1.0 MCF) bacterial suspension, we added 2–5 mg of bacterial biomass from the growth medium and diluted it with physiological saline. Then, aliquots of the 1 mg/mL bacterial suspension were distributed to each well of the plates and incubated at 37°C with 5% $CO_2$ for 10–21 days. H37Rv (ATCC 27294) served as a pan-susceptible control in each batch. Based on CLSI M62, the breakpoints for all drugs were set at 0.2 μg/ml for INH, 0.5 μg/ml for FLQ, 1.0 μg/ml for AMK, 5.0 μg/ml for KAN, 2.0 μg/ml for CAP, and 5.0 μg/ml for ETH.

### Discrepant result resolution (sequencing)

Discrepant results were defined as specimens where the Xpert MTB/XDR test and pDST results were not in agreement. These discrepancies were categorized as false-positive or false-negative results. False-positive results refer to a scenario where the Xpert MTB/XDR result indicates drug resistance, while the pDST result shows susceptibility. In addition, a false-negative result is the opposite. Specimens with discrepant results were investigated by Sanger sequencing. Sanger sequencing was performed on INH (katG, inhA promoter, fabG1and oxyR-ahpC intergenic region genes) and FLQ (gyrA and gyrB genes). The primers were derived from previously published articles [10,11], and their sequences are shown in S1 Table. The PCR assays were accomplished using the Taq Pro HS DNA Polymerase kit (Vazyme) and the ArtGene (TM) A300 device (LongGene). The PCR assay was performed under the following conditions: 95°C for 30 s, 45 cycles at 95°C for 30 s, 60°C for 30 s, and 72°C for 60 s, and each run contained a negative control (water). The PCR products were sent to Sangon Biotech for Sanger sequencing.

### Statistical data analysis

The clinical performance of the Xpert MTB/XDR test for resistance detection of INH, FLQ, AMK, KAN, CAP, and ETH was evaluated using the following outcome measures: sensitivity, specificity, positive predictive value (PPV), and negative predictive value (NPV) compared with pDST as a phenotypic reference standard.

SPSS Statistics software (version 21, IBM, NY, USA) was used to perform the statistical analysis. The results of the pDST and MTB/XDR tests were analyzed using Kappa and McNemar's tests, and a value of $P < 0.05$ was considered statistically significant.

Sanger sequencing, as a molecular reference standard, can be integrated with pDST to re-evaluate the antimicrobial susceptibility and resistance profiles in samples. The composite reference test (pDST+sequencing) results were categorized as "Resistant" if either pDST or sequencing results were "Resistant", and "Susceptible" if both pDST and sequencing were "Susceptible". Following that, the clinical performance of the MTB/XDR test was also evaluated using the composite reference test (pDST + sequencing), as described above.

## Results

### Specimen characteristics

As shown in Table 1, the final data analysis was performed on 276 specimens, of which 224 (81.1%) were sputum specimens, 45 (16.3%) were lavage fluids, 5 (1.8%) were pus, 1 (0.4%) was puncture fluid, and 1 (0.4%) was feces. Among

**Table 1. Demographics and diagnostic characteristics of specimens included in the analysis.**

| Parameter | Overall (N=276) |
|---|---|
| Age (years) | |
| median | 52 |
| Range | 17-88 |
| Sex | |
| Female | 74 (26.8%) |
| Male | 202 (73.2%) |
| Sample type (Patient type) | |
| Sputum (Pulmonary tuberculosis) | 224 (81.1%) |
| Lavage fluid (Pulmonary tuberculosis) | 45 (16.3%) |
| Pus (Lymph Node tuberculosis) | 5 (1.8%) |
| Puncture fluid (Bone tuberculosis) | 1 (0.4%) |
| Feces (Intestinal tuberculosis) | 1 (0.4%) |
| Diagnostic characteristics | |
| MGIT culture/ AFB smear | |
| % Culture positive [n/N] | 100% (276/276) |
| % Smear positive [n/N] | 57.6% (159/276) |
| % Smear negative [n/N] | 42.4% (117/276) |
| Ultra | |
| % positive (MTB detected) [n/N] | 97.5% (269/276) |
| % RIF sensitive [n/N] | 25.3% (68/269) |
| % RIF resistant [n/N] | 74.7% (201/269) |
| % negative [n/N] | 2.5% (7/276) |

Nevertheless, the sample type is limited, and clinical samples sizes are tiny; a more comprehensive evaluation of the Xpert MTB/XDR test with larger and more diversified samples is required later.

these, 269 samples were collected from pulmonary tuberculosis patients; 7 samples were derived from extrapulmonary tuberculosis, including 5 lymph node tuberculosis (pus samples), 1 bone tuberculosis (puncture fluid sample), and 1 intestinal tuberculosis (feces sample). Sputum and Lavage fluid were the most common sample types. Among them, the median age was 52 years (17–88), with men accounting for 73.2%. Of 276 samples, 276 (100%) were positive for culture and 159 (57.6%) were positive for smear. In addition, 97.5% (269/276) of the specimens tested positive on the Ultra test, with 74.7% (201/269) being RIF resistant and 25.3% (68/269) being RIF sensitive.

### Diagnostic performance of the Xpert MTB/XDR test for drug resistance prediction

The diagnostic performance of the Xpert MTB/XDR test against pDST used for all drugs is shown in Table 2. Using 276 samples, the Xpert MTB/XDR test detected INH and FLQ resistance with a sensitivity of 95.77% (95% CI: 91.83–98.16) and 93.83% (95% CI: 86.18–97.97), respectively. In addition, the sensitivity for AMK, KAN, CAP, and ETH detection was each less than 90%: 73.33% (95% CI: 44.90–92.21), 73.33% (95% CI: 44.90–92.21), 62.50% (95% CI: 24.49–91.48), and 66.67% (95% CI: 22.28–95.67) respectively. In contrast, all drugs demonstrated a detection specificity better than 90%.

A total of 11 discrepant samples with 3 false-positive and 8 false-negative results were identified between the Xpert MTB/XDR test and pDST in detecting INH resistance. Similarly, 22 samples detecting FLQ resistance were either 17 false-positive or 5 false-negative. All samples above were retested by sequencing, which identified mutations predicting INH and FLQ resistance, determining whether 33 samples were resistant by combining pDST and sequencing results. As shown in Table 2, both sensitivity and specificity in detecting resistance to INH and FLQ using the composite test results

**Table 2. Diagnostic accuracy of Xpert MTB/XDR test with pDST and composite reference test (pDST+sequencing) results.**

| Drug | No. Of. Samples | TP | FN | TN | FP | Performance characteristics Sensitivity (% [95% CI]) | Specificity (% [95% CI]) | PPV (% [95% CI]) | NPV (% [95% CI]) | Accuracy (% [95% CI]) | Kappa |
|---|---|---|---|---|---|---|---|---|---|---|---|
| pDST | | | | | | | | | | | |
| INH | 276 | 181 | 8 | 84 | 3 | 95.77 (91.83-98.16) | 96.55 (90.25-99.28) | 98.37 (95.20-99.46) | 91.30 (84.19-95.39) | 96.01 (92.98-97.99) | 0.91 |
| FLQ | 276 | 76 | 5 | 178 | 17 | 93.83 (86.18-97.97) | 91.28 (86.41-94.84) | 81.72 (73.88-87.60) | 97.27 (93.83-98.81) | 92.03 (88.18-94.94) | 0.82 |
| AMK | 144 | 11 | 4 | 127 | 2 | 73.33 (44.90-92.21) | 98.45 (94.51-99.81) | 84.62 (57.35-95.74) | 96.95 (93.20-98.66) | 95.83 (91.15-98.46) | 0.76 |
| KAN | 144 | 11 | 4 | 123 | 6 | 73.33 (44.90-92.21) | 95.35 (90.15-98.27) | 64.71 (44.21-80.92) | 96.85 (92.99-98.62) | 93.06 (87.60-96.62) | 0.65 |
| CAP | 144 | 5 | 3 | 128 | 8 | 62.50 (24.49-91.48) | 94.12 (88.74-97.43) | 38.46 (20.91-59.63) | 97.71 (94.57-99.05) | 92.36 (86.74-96.13) | 0.44 |
| ETH | 144 | 4 | 2 | 125 | 13 | 66.67 (22.28-95.67) | 90.58 (84.43-94.89) | 23.53 (12.51-39.84) | 98.43 (95.27-99.49) | 89.58 (83.40-94.05) | 0.31 |
| pDST+sequencing | | | | | | | | | | | |
| INH | 276 | 183 | 5 | 87 | 1 | 97.34 (93.90-99.13) | 98.86 (93.83-99.97) | 99.46 (96.30-99.92) | 94.57 (87.99-97.64) | 97.83 (95.33-99.20) | 0.95 |
| FLQ | 276 | 93 | 5 | 178 | 0 | 94.90 (88.49-98.32) | 100 (97.95-100) | 100 (96.11-100) | 97.27 (93.81-98.82) | 98.19 (95.82-99.41) | 0.96 |

INH, Isoniazid; FLQ, Fluoroquinolone; AMK, amikacin; KAN, kanamycin; CAP, capreomycin; ETH, ethionamide; TP, true-positive results; FN, false-negative results; TN, true-negative results; FP, false-positive results; PPV, Positive Predictive Value; NPV, Negative Predictive Value; CI, confidence interval.

(pDST+sequencing) as the reference have improved, with FLQ detection specificity reaching 100% (95% CI: 97.95−100). The Sanger sequencing results of 33 discrepant samples are shown in Table 3. Among the 11 samples detecting INH resistance, 7 were positive with mutation types including katG S315T, inhA-15 C→T, inhA16 A→C, and ahpC-32 G→A, and katG S315T was the most. In addition, 22 samples detecting FLQ were gyrA gene mutations, including codons 89, 90, 91, and 94, with gyrA A90V and gyrA S95T being the most common.

## Discussion

The MTBDR*plus* and MTBDR*sl* version 2.0 (HAIN Life Sciences, Germany) assays are currently recommended by the WHO as molecular tests to detect INH and RIF resistance and FLQ and SLID resistance, respectively [12]. However, both tests require an independent and continuous process PCR amplification, reverse hybridization, and chromogenic reaction, which is time-consuming (MTBDR*sl* >6h). Furthermore, both tests must be performed in a Biosafety Level 2 laboratory, with people receiving professional training to reduce contamination risks [13]. Considering the foregoing, the promotion of this platform in China has not been smooth. In comparison, the Xpert MTB/XDR test generated molecular DST results in less than 90 minutes, with limited technical and infrastructural requirements, making it more suitable for near-patient or point-of-care testing.

All of these specimen types (sputum, lavage fluid, pus, puncture fluid, and feces) can be used in this study to successfully detect related drug resistance using the Xpert MTB/XDR test. When sputum samples are unavailable, for patients with extrapulmonary tuberculosis, different types of specimens can be used for Xpert testing. Therefore, this is critical for the timely detection, treatment, prevention, and control of extrapulmonary tuberculosis.

**Table 3. Sanger sequencing results of 33 discrepant samples in detecting INH and FLQ resistance.**

| Drug | pDST | Xpert MTB/XDR | Sanger sequencing | Number of samples | Total number of samples |
|------|------|---------------|-------------------|-------------------|-------------------------|
| INH | R | S | katG S315T | 3 | 11 |
| | R | S | inhA16 A→C | 1 | |
| | R | S | ahpC-32 G→A | 1 | |
| | R | S | S | 3 | |
| | S | R | katG S315T | 1 | |
| | S | R | inhA-15 C→T | 1 | |
| | S | R | S | 1 | |
| FLQ | R | S | gyrA S95T | 4 | 22 |
| | R | S | gyrA D94G; gyrA S95T | 1 | |
| | S | R | gyrA A90V; gyrA S95T | 5 | |
| | S | R | gyrA S95T | 4 | |
| | S | R | gyrA S91P; gyrA S95T | 3 | |
| | S | R | gyrA D94A; gyrA S95T | 2 | |
| | S | R | gyrA D89N; gyrA S95T | 1 | |
| | S | R | gyrA D94N; gyrA S95T | 1 | |
| | S | R | gyrA D94Y; gyrA S95T | 1 | |

INH, Isoniazid; FLQ, Fluoroquinolone; R, resistant; S, sensitive.

The current study's clinical performance of the Xpert MTB/XDR test is similar to previous studies, with a sensitivity of >90% for INH and FLQ compared to pDST, meeting the minimum criteria for the target product profile for next-generation DST assays [14,15]. This study found that the Xpert MTB XDR test had a lesser sensitivity (<75%) in predicting SLID resistance compared to pDST. The best result was that the specificity for detecting all drugs exceeded 90%. When considering pDST + sequencing, the sensitivity and specificity of the Xpert MTB/XDR assay for INH and FLQ drug targets increased, especially the detection specificity of FLQ has reached 100% (95% CI: 97.95–100). As shown in Table 3, the sequencing analysis restrictions hampered the resolution of the inconsistencies detected between the pDST and the Xpert MTB/XDR test [16,17]. Among the eight false negatives for INH resistance, five had mutations, primarily katG S315T, as identified by Sanger sequencing. This condition could be caused by a mixed infection containing both drug-resistant and drug-sensitive strains. The pDST may detect resistance due to the growth of the resistant subpopulation, but Xpert MTB/XDR might yield a "false-sensitive" result if the sensitive strains predominate. However, Sanger sequencing can detect drug-resistant mutations through in-depth analysis. All 17 false positives for FLQ resistance were attributed to gyrA mutations identified by Sanger sequencing. The reason for this is that the pDST may still be sensitive at the critical concentration, but Xpert MTB/XDR can identify the relevant gene mutations [18]. The enhanced sensitivities demonstrate the assay's accuracy in predicting resistance, particularly for INH and FLQ. These performance attributes make the Xpert MTB/XDR test ideal for reducing the current challenges of obtaining INH and FLQ DST results for better DR-TB patient management.

The sensitivity of second-line drugs remains insufficient, similar to or slightly lower than the results reported in other reports [19,20]. The sensitivity and specificity of the second-line drugs were 73.33% (95% CI: 44.90–92.21) and 98.45% (95% CI: 94.51–99.81) of AMK, 73.33% (95% CI: 44.90–92.21) and 95.35% (95% CI: 90.15–98.27) of KAN, 62.50% (95% CI: 24.49–91.48) and 94.12% (95% CI: 88.74–97.43) of CAP, 66.67% (95% CI: 22.28–95.67) and 90.58% (95% CI: 84.43–94.89) of ETH, respectively. The high specificity of the Xpert MTB/XDR test reflects the ability to determine drug resistance using highly specific probes of crucial mutation sites. However, its sensitivity to SLIDs is limited by incomplete coverage or unclear multigene resistance mechanisms [21]. The resistance of aminoglycoside antibiotics like AMK and

KAN is often linked to rrs gene mutations. However, the Xpert MTB/XDR assay only detects partial mutation sites (e.g., rrs A1401G) and ignores low-frequency resistance mutations like C517T, lowering sensitivity. Similarly, CAP resistance can involve both rrs and tlyA genes, but the Xpert MTB/XDR only detects the rrs gene while omitting tlyA mutations risks missed diagnoses. The ETH resistance commonly involves inhA promoter regions and ethA genes, causing a decrease in Xpert MTB/XDR sensitivity without covering ethA mutations. Additionally, resistance mechanisms for these drugs may involve multiple genes/mutation sites not fully captured by Xpert MTB/XDR, leading to undetected resistance mutations. Our team is continuously researching the resistance mechanisms of second-line drugs with the goal of achieving more precise molecular detection of tuberculosis drug resistance.

As a next-generation molecular diagnostic tool, the Xpert MTB/XDR significantly improves the diagnosis and treatment of drug-resistant tuberculosis by rapidly detecting MTB and its resistance to multiple first- and second-line drugs [22,23]. The Xpert MTB/XDR test enables precise drug selection for TB patients: if a katG mutation (high-level resistance) is detected, INH should be discontinued; if an inhA mutation (low-level resistance) is identified, high-dose INH (10–15 mg/kg) may be retained. For FLQ resistance, moxifloxacin and levofloxacin should be avoided in favor of bedaquiline or linezolid. For XDR-TB patients, direct identification of resistance to aminoglycosides (e.g., AMK) and ETH prevents ineffective drug combinations and optimizes treatment strategies. Xpert MTB/XDR can prevent the emergence of resistant strains by shortening diagnostic time, lowering community transmission risks, and rapidly terminating ineffective drugs in drug-resistant TB patients. In resource-limited settings, it serves as a one-stop inspection tool, replacing multi-step conventional detection workflows.

However, this study has a few limitations. Firstly, different specimen types and more specimens from suspected tuberculosis patients are needed to further verify the Xpert MTB/XDR test. Secondly, specimens inconsistent between the Xpert MTB/XDR test and pDST in detecting SLIDs resistance did not undergo further Sanger sequencing.

Thirdly, the lack of true-positive SLID resistance samples may have limited our confidence in the accurate evaluation of sensitivity for SLID drugs, and we plan to include more SLID resistance samples in the future to further analyze this test.

This study shows that the Xpert MTB/XDR test has high sensitivity and specificity compared to pDST, especially in INH and FLQs. The Xpert MTB/XDR significantly improves clinical decision-making and treatment outcomes for drug-resistant tuberculosis through rapid, precise drug susceptibility testing, serving as a critical tool for achieving personalized treatment and TB control. Future advancements should prioritize cost reduction and expanding coverage of drug resistance gene panels to enhance accessibility for broader patient populations.

## Supporting information

**S1 Table. Primers of Sanger sequencing for detecting INH and FLQ resistance sites.**
(DOCX)

**S1 Data. Data set.**
(XLSX)

## Acknowledgments

We gratefully acknowledge the assistance from Anhui Chest Hospital, and Anhui Provincial Center for Disease Control and Prevention. We thank Xinghui Gao and Yi-Wei Tang, who are employees of Danaher/Cepheid, the commercial manufacturer of the Xpert MTB/XDR assay.

## Author contributions

**Formal analysis:** Bing Zhao, Hui Xia, Ruida Xing.

**Investigation:** Ruiqing Zhang, Chong Teng.

**Resources:** Xundi Bao, Fangjin Bao, Dongfang Xu, Zhou Liu, Yue Li.

**Writing – original draft:** Ruiqing Zhang.

**Writing – review & editing:** Xichao Ou, Yanlin Zhao.

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
