## [Decision Letter · Decision Letter 0]

26 Jun 2025

Dear Dr. zhang,

Thank you for submitting your manuscript to PLOS ONE. After careful consideration, we feel that it has merit but does not fully meet PLOS ONE’s publication criteria as it currently stands. Therefore, we invite you to submit a revised version of the manuscript that addresses the points raised during the review process.

We look forward to receiving your revised manuscript.

Kind regards,

Sarman Singh, MD, FRSC, FRCP

Academic Editor

PLOS ONE

Journal Requirements:

4. Please remove all personal information, ensure that the data shared are in accordance with participant consent, and re-upload a fully anonymized data set.

5. We notice that your supplementary [figures/tables] are included in the manuscript file. Please remove them and upload them with the file type 'Supporting Information'. Please ensure that each Supporting Information file has a legend listed in the manuscript after the references list.

Additional Editor Comments :

1. In addition to suggestions from the reviewers, the authors are requested to improvise the English language requires a significant improvisation. For example but not limited to "xxx relative to pDST" should be changed to "xxx compared to pDST".

2. Also the refences are not in the proper formatting. "Organization WH" should be written as World Health Organization" or "WHO". There are many more grammatical errors.

Reviewers' comments:

Reviewer's Responses to Questions

**Comments to the Author**

1. Is the manuscript technically sound, and do the data support the conclusions?

Reviewer #1: Yes

Reviewer #2: Yes

2. Has the statistical analysis been performed appropriately and rigorously?

Reviewer #1: Yes

Reviewer #2: Yes

3. Have the authors made all data underlying the findings in their manuscript fully available?

Reviewer #1: Yes

Reviewer #2: Yes

4. Is the manuscript presented in an intelligible fashion and written in standard English?

Reviewer #1: Yes

Reviewer #2: Yes

Reviewer #1: 1. In the results section, line 203 ”A total of 11 discrepant samples with 3 false-positive and 8 false-negative...."

I think this false negative result needs to be discussed in the discussion section.

2. In the results section, line 205 "17 false-positive or 5 false-negative..."

I think this false positive result needs to be discussed in the discussion section.

3. What criteria are used to determine that someone is suspected of having MDR-TB in the methods section?

Reviewer #2: Thank you for the manuscript. It was well-written and it is well-appreciated. From my side only few minor comments:

- Line 66-68: I would suggest to use updated information in the introduction part.

- Line 108-112: Clinically suspected MDR patients? The mentioned sentences are not clear, for instance what are the chances for a patient that would be smear positive and but clinically shows improvement?

- Line 254-255: “The sensitivity of second-line drugs remains insufficient, similar to or slightly lower than the results reported in other reports” Ref: [18-19]

**Do you want your identity to be public for this peer review?** For information about this choice, including consent withdrawal, please see our Privacy Policy

Reviewer #1: **Yes: ** Tutik Kusmiati

Reviewer #2: **Yes: ** Ahmad Reza Yosofi

---

## [Author Response · Author response to Decision Letter 1]

11 Aug 2025

Dear Editor-in-Chief, Editor and Reviewer,

Manuscript Number: PONE-D-25-27893R1

Thanks to the editor for arranging the review. Additionally, we would like to thank the editor and reviewers for their valuable comments on our manuscript. We have carefully checked the manuscript and answered the reviewer's questions point by point. The revised parts are marked in red in the revised manuscript.

We hoped you will find our revised manuscript suitable for publication in PLOS One.

Additional Editor Comments :

1.In addition to suggestions from the reviewers, the authors are requested to improvise the English language requires a significant improvisation. For example but not limited to "xxx relative to pDST" should be changed to "xxx compared to pDST".

Answer: We have thoroughly revised the manuscript to improve the English language, including the specific example mentioned.

2. Also the refences are not in the proper formatting. "Organization WH" should be written as World Health Organization" or "WHO". There are many more grammatical errors.

Answer: We have reformatted all references to ensure they adhere to the Plos One style, including correcting "Organization WH" to "World Health Organization" or "WHO" as appropriate. Additionally, we have conducted a comprehensive review of the text to correct grammatical errors throughout the manuscript.

Review Comments to the Author

Reviewer #1:

1. In the results section, line 203 ”A total of 11 discrepant samples with 3 false-positive and 8 false-negative...."

I think this false negative result needs to be discussed in the discussion section.

Answer: We have added a discussion regarding the false negative results in the discussion section (line 278-283). We elaborated on the potential causes of these false positives in the context of our findings.

2.In the results section, line 205 "17 false-positive or 5 false-negative..."

I think this false positive result needs to be discussed in the discussion section.

Answer: We have included a discussion on the false positive results in the discussion section (line 283-287). We elaborated on the potential causes of these false positives in the context of our findings.

3.What criteria are used to determine that someone is suspected of having MDR-TB in the methods section?

Answer: We have clarified the criteria used to determine suspicion of MDR-TB in the methods section. This includes specific clinical, radiological, and microbiological parameters that guide the suspicion of MDR-TB in patients. (line 107-111)

Reviewer #2:

Thank you for the manuscript. It was well-written and it is well-appreciated. From my side only few minor comments:

- Line 66-68: I would suggest to use updated information in the introduction part.

Answer: We have revised the introduction to include data of China in 2023 according to Global tuberculosis report 2024 (line 66-68).

- Line 108-112: Clinically suspected MDR patients? The mentioned sentences are not clear, for instance what are the chances for a patient that would be smear positive and but clinically shows improvement?

Answer: We have clarified the criteria used to determine suspicion of MDR-TB in the methods section (line 107-111). Regarding what are the chances for a patient that would be smear positive and but clinically shows improvement, we have not conducted any statistical analysis, and further research will be carried out subsequently.

- Line 254-255: “The sensitivity of second-line drugs remains insufficient, similar to or slightly lower than the results reported in other reports” Ref: [18-19]

Answer: We have already made modifications in the original manuscript.

Kind regards.

Yanlin Zhao, PhD

National Key Laboratory of Intelligent Tracking and Forecasting for Infectious Diseases, National Center for Tuberculosis Control and Prevention, Chinese Centre for Disease Control and Prevention, No. 155, Changbai Street, Changping District, Beijing 102206, China

E-mail address: zhaoyl@chinacdc.cn

---

## [Editor Report · Decision Letter 1]

14 Aug 2025

Evaluation of the Xpert MTB/XDR test for detection of Isoniazid, Fluoroquinolones, and second-line injectable drugs resistance to Mycobacterium tuberculosis—Anhui Province, China

PONE-D-25-27893R1

Dear Dr. zhang,

We’re pleased to inform you that your manuscript has been judged scientifically suitable for publication and will be formally accepted for publication once it meets all outstanding technical requirements.

Kind regards,

Sarman Singh, MD, FRSC, FRCP

Academic Editor

PLOS ONE
---

## [Editor Report · Acceptance letter]

PONE-D-25-27893R1

PLOS ONE

Dear Dr. Zhang,

I'm pleased to inform you that your manuscript has been deemed suitable for publication in PLOS ONE. Congratulations! Your manuscript is now being handed over to our production team.

Kind regards,

on behalf of

Professor Sarman Singh

Academic Editor

PLOS ONE